

# Digital marketing program design based on abnormal consumer behavior data classification and improved homomorphic encryption algorithm

Jun Cui[1], Hao Jiang[2] and Zhendan Xu[3]

[1] Business School, Hohai University, Nanjing, Jiangsu, China
[2] Southwest Jiaotong University Hope College, Cheng Du, Si Chuan, China
[3] School of Management, Zhejiang University, Hangzhou, Zhejiang, China

## ABSTRACT

This article endeavors to delve into the conceptualization of a digital marketing framework grounded in consumer data and homomorphic encryption. The methodology entails employing GridSearch to harmonize and store the leaf nodes acquired post-training of the CatBoost model. These leaf node data subsequently serve as inputs for the radial basis function (RBF) layer, facilitating the mapping of leaf nodes into the hidden layer space. This sequential process culminates in the classification of user online consumption data within the output layer. Furthermore, an enhancement is introduced to the conventional homomorphic encryption algorithm, bolstering privacy preservation throughout the processing of consumption data. This augmentation broadens the applicability of homomorphic encryption to encompass rational numbers. The integration of the Chinese Remainder Theorem is instrumental in the decryption of consumption-related information. Empirical findings unveil the exceptional generalization performance of the amalgamated model, exemplifying an AUC (area under the curve) value of 0.66, a classification accuracy of 98.56% for online consumption data, and an F1-score of 98.41. The enhanced homomorphic encryption algorithm boasts attributes of stability, security, and efficiency, thus fortifying our proposed solution in facilitating companies' access to precise, real-time market insights. Consequently, this aids in the optimization of digital marketing strategies and enables pinpoint positioning within the target market.

# INTRODUCTION

Consumer behavioral data is represented as a time-series-based univariate sequence, where each point in the sequence corresponds to a specific behavior (*Singh & Yassine, 2018*). This data is sourced from internal company applications, primarily derived from commercial applications like recommendations, ad placement, online financial services, and network security, and is generated by operations and maintenance personnel. Online consumer

Corresponding author
Zhendan Xu, 22120442@zju.edu.cn

behavioral data is crucial as it reflects users' consistent website visits within a defined timeframe. Analyzing this data enables companies to refine their business strategies for different user groups, underscoring the importance of detecting and classifying abnormal consumer behavioral patterns (*Fan, Jiang & Lin, 2022*). Anomaly detection involves establishing a database of users' normal behavioral features and comparing their current behaviors against this database. If the deviation is substantial enough, an anomaly is detected. The purpose is to identify outliers or patterns that significantly differ from the majority of the data. Analyzing consistent online consumer behavioral data helps companies identify general behavior trends, using this temporal regularity to predict user clicking and purchasing tendencies (*Liu, He & Han, 2019*). By cross-referencing behavior feedback, it's possible to pinpoint users with irregular time series, offering valuable insights to refine system recommendations. Abnormal online consumer behavioral data encompasses disjointed, missing, or disordered sequences resulting from issues in users' ongoing behavior records (*Wu, Shi & Lin, 2020*). The study of online consumer behavioral data has distinct characteristics. Despite the large data volume, instances of abnormal behavior are exceedingly rare, making them challenging to identify. Data formats are often complex, involving high-dimensional classification IDs that can be sparse and intricate, frequently remaining consistent over extended periods. Additionally, due to the dynamic nature of real-world business operations, encountered exceptions vary widely, posing challenges for existing anomaly detection and classification methods.

In the era of digital advancements, the collection and analysis of consumer data have become incredibly important for companies. However, ensuring privacy protection and data security has become a critical concern (*Saura, Ribeiro-Soriano & Palacios-Marqués, 2021*; *Line, Dogru & El-Manstrly, 2020*). Especially in the field of digital marketing, advanced encryption technologies play a pivotal role in safeguarding data privacy and security. By employing homomorphic encryption algorithms, consumer data can be encrypted and analyzed while remaining in an encrypted state. This approach allows companies to derive valuable insights into consumer behavior and preferences without exposing the original data (*Iezzi, 2020*). Homomorphic encryption is a specialized data encryption technique that enables data processing without the need to access the actual data itself. Homomorphic encryption is a cryptographic technique that holds paramount significance in the realm of secure data processing and privacy preservation. At its core, this encryption algorithm operates by encrypting data using both public and private keys, yielding a unique form of ciphertext that retains its mathematical properties even when subjected to computations. The distinguishing feature of homomorphic encryption lies in its ability to maintain consistency between the outcome of decrypting the ciphertext and the result of performing computations directly on the plaintext data post-encryption and subsequent decryption, as highlighted by reference (*Hamza, Hassan & Ali, 2022*). This property opens up a realm of possibilities, enabling collaborative computation among multiple sources while ensuring data privacy. Homomorphic encryption significantly mitigates the risk of private data leakage, a critical concern in today's data-centric world. Furthermore, it plays a pivotal role in safeguarding data sharing by preserving privacy, bolstering the confidence of individuals and organizations alike in sharing sensitive

information securely. In summary, homomorphic encryption stands as a cornerstone in the field of data security, offering robust protection, privacy preservation, and collaborative computation capabilities.

Given that research on privacy protection in machine learning is still in its early stages, specific methods for ensuring privacy have been proposed for various application scenarios. Techniques like differential privacy and homomorphic encryption multiparty computation have been explored within the context of machine learning. However, challenges persist, particularly concerning the efficiency and security assessment criteria for machine learning systems (*Xu, Sun & Cardell-Oliver, 2023*; *Altaee & Alanezi, 2021*). Consequently, proposing a privacy-preserving model tailored to consumer data scenarios becomes both scientifically and practically significant. Such a model would fulfill the requirements of data privacy, security, and regulations, enabling cost-effective, efficient, and accurate utilization of data from different sources. This, in turn, would empower the creation of well-founded marketing strategies based on an extensive pool of consumer data.

The primary contributions of this study are as follows:

1. The design of an online consumer anomaly behavior data classification method based on CatBoost-RBF. This method employs a stacked autoencoder structure with two layers of encoders, thus mitigating overfitting when calculating leaf nodes and ensuring the utilization of the entire dataset for learning. It effectively addresses predictive problems.

2. Enhancement of conventional homomorphic encryption algorithms to optimize privacy protection in consumer data handling. It extends the scope of homomorphic encryption to rational numbers and introduces the Chinese Remainder Theorem for decrypting consumer information.

## RELATED WORKS

In the realm of online consumer behavioral data, a multitude of data sources exist, each characterized by its own intricate and constantly evolving scenarios. Detecting anomalies within this data presents challenges due to the rarity of anomalous samples and the inherent difficulty in timely detection within real-world application environments. As a result, researchers around the world have introduced a plethora of algorithms designed to detect and classify anomalies in online consumer behavioral data. The objective is to improve the precision of anomaly detection and classification methods.

*Mirzaei, Nikpour & Nezamabadi-pour (2021)* proposed a hybrid approach that combines the density concept and clustering techniques to address the classification problem in imbalanced datasets. They used K-means clustering to obtain sample densities and achieved promising results by applying support vector machine (SVM) for classification across multiple datasets. *Korkmaz (2020)* investigated the issue of imbalanced data classification and employed a deep neural network (DNN) to classify the dataset. Various undersampling and oversampling methods were employed to balance the dataset and improve classification performance. *Zhou, Lu & Liu (2023)* introduced a fuzzy decision tree algorithm that leverages clustering and HFS-based fuzzy decision trees. This method demonstrated high

AUC values when applied to multiple datasets. *Yu, Zhou & Tang (2018)* proposed a deep belief network (DBN)-based resampling SVM integrated learning model to address data imbalance in credit risk assessment. By using SVM as the base classifier and DBN for integration, they incorporated income-sensitive category weights, resulting in improved accuracy for credit risk classification. *Chen, Li & Su (2019)* presented an AdaBoost-KNN integration algorithm to tackle the multi-class imbalanced data classification problem. This method utilized K-nearest neighbor (KNN) as the base classifier and exhibited high classification accuracy on unbalanced datasets, as evaluated by various metrics. *Tanha, Abdi & Samadi (2020)* compared the performance of binary and multiclassification boosting algorithms on various multiclass datasets. They found that the Catboost algorithm outperformed other boosting algorithms significantly, particularly on large multiclass unbalanced datasets. *Nagabushanam, Jayan & Joel (2021)* addressed the imbalanced data classification problem by adding different layers of Cost Criteria function to the output layer of a convolutional neural network (CNN). They experimentally verified the effectiveness of this approach on various classification datasets. *Jhaveri, Khedkar & Kantharia (2019)* conducted an analysis of advertisement data from the Kickstarter platform. They compared the performance of four algorithms, namely Random Forest, Catboost, XGBoost, and Adaboost, to predict ad effectiveness. Their findings revealed that a combination of Random Forest and Adaboost yielded the best classification results.

Encrypted machine learning involves encrypting data and performing various machine learning tasks, such as classification and clustering, on ciphertext. The Faster-CryptoNets scheme proposed by *Chou, Beal & Levy (2018)* optimizes the model simplification process by combining neural network pruning techniques to reduce the number of parameters in the original model, thereby reducing the number of multiplication operations. This optimization leads to shorter inference times but may result in a loss of model accuracy. Additionally, the ReLU function is replaced with a low-order polynomial approximation for maximum sparse coding. Compared to the original CryptoNets scheme, the Faster-CryptoNets scheme improves prediction rates by approximately 10 times.

To achieve privacy-preserving inference for complex neural networks, *Lee, Kang & Lee (2022)* utilize a low-order polynomial approximation for activation function computation and construct a secure batch normalization layer based on fully homomorphic encryption. This approach enables the neural network model to be suitable for more complex privacy-preserving classification tasks. While the above schemes primarily focus on the secure inference process of neural network models, when users outsource the model training task to a cloud server, there arises a concern regarding data privacy leakage during the training process. In this context, some scholars have applied homomorphic encryption schemes to neural network model training. *Pulido-Gaytan, Tchernykh & Cortés-Mendoza (2021)*; *Zhang, Wang & Wu (2018)* propose the GELU-Net scheme, which uses Paillier semi-homomorphic encryption to avoid accuracy loss caused by polynomial approximation of the activation function and circumvent multiplicative homomorphism between ciphertexts. *Han & Ki (2020)* improve the bootstrapping process of homomorphic encryption schemes to reduce the secure computation time of neural networks, albeit with a slight decrease in accuracy. *Chen, Chillotti & Song (2019)* employ Chebyshev polynomials to approximate

nonlinear functions, enhancing the accuracy of the ReLU activation function to some extent. This scheme utilizes full homomorphic encryption to encrypt data and sets a noise threshold that requires the server to return the encryption model to the user for decryption when the noise of the multiplication operation reaches this threshold.

Currently, the predominant method for implementing cryptographic machine learning is centered around the utilization of homomorphic encryption schemes. Nevertheless, the efficiency of homomorphic encryption is contingent on the nature and quantity of operations involved, prompting researchers to explore alternative avenues, such as multi-party computation and algorithm approximation (*Hei, Liang & Wang, 2020*). Multi-party computation involves collaborative operations among numerous participants, which can tackle the intricacies of complex ciphertext operations. However, the creation of transmission protocols between multiple parties can be intricate, potentially affecting processing efficiency. Furthermore, participants must adhere to the protocol concurrently to ensure accuracy and security, introducing potential reliability concerns in comparison to single-party computation. Conversely, algorithm approximation circumvents the need for protocol development. Nevertheless, to avoid convoluted operations, the original algorithm must be rewritten, potentially impacting the precision of results to some degree (*Duan, 2022*). Although fully homomorphic encryption theoretically permits arbitrary computations, contemporary schemes come with certain limitations. For instance, they might exclusively support integer-type data, necessitate a predetermined multiplication depth that confines the count of addition and multiplication operations, and lack functionality for operations such as comparisons and identifying maximum values.

## MODEL DESIGN

This framework comprises four key entities: the owner of consumption data, the user of consumption data, a third-party computing center, and an advertising operator. In order to uphold the confidentiality and integrity of data, the e-commerce entity employs encryption techniques to encrypt the consumption data within the system. Subsequently, this encrypted data is securely stored in a protected storage location. The third-party computing center then retrieves the ciphertext from the storage location for further processing. When a merchant intends to dispatch targeted marketing SMS messages, they provide input that includes desired customer categories and corresponding marketing content. Thanks to the implementation of privacy-preserving measures, merchants are unable to access individual personal information for the purpose of personalized marketing. This protective measure ensures consumer privacy while still facilitating the precise targeting of marketing strategies toward specific consumer groups.

The encryption of consumption data plays a crucial role in ensuring the protection of sensitive information, benefiting both consumers and businesses. Furthermore, the involvement of a third-party computing center facilitates the efficient retrieval and processing of encrypted data, relieving individual merchants from excessive workload. Additionally, the separation of personal information from marketing activities serves as a safeguard against potential privacy breaches, fostering consumer trust in e-commerce

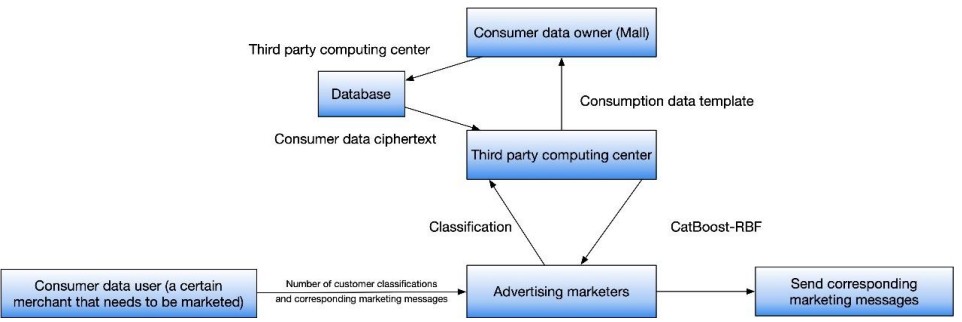

**Figure 1** **Overall framework of the proposed model.** Overall, this scheme provides a strong framework for secure data handling and effective marketing in the e-commerce sector, enabling targeted marketing campaigns while upholding consumer privacy rights.

platforms. Overall, this scheme provides a strong framework for secure data handling and effective marketing in the e-commerce sector, enabling targeted marketing campaigns while upholding consumer privacy rights. As shown in Fig. 1.

## Classification of consumption data based on improved Catboost

In order to effectively classify online consumer abnormal behavior data, this study employs a CatBoost-RBF-based method. CatBoost is utilized to address the primary challenge of enhancing the efficiency and accuracy of the classification function. By avoiding overfitting during the calculation of leaf nodes, CatBoost improves the algorithm's generalization capability and ensures effective learning and prediction on the entire dataset (*Hancock & Khoshgoftaar, 2020*). The sample training set undergoes testing by randomly shuffling the sample data features and converting classification features into data value features, as illustrated in Eq. (1):

$$x_{i,k} = \frac{\sum_{j=1}^{p-1} \left[ x_{\sigma_{j,k}} = x_{\sigma_{p,k}} \right] \cdot Y_j + a \cdot p}{\sum_{j=1}^{p-1} \left[ x_{\sigma_{j,k}} = x_{\sigma_{p,k}} \right] + a} \tag{1}$$

where p denotes the average target value in the dataset, a denotes the weighting factor. $x_{\sigma_{j,k}}, x_{\sigma_{p,k}}$ and $Y_j$ denote the average target value in the training sample k of $\sigma_j - $ th and $\sigma_p - $ th feature. The CatBoost algorithm utilizes a symmetric decision tree structure and transforms processable class features into processed label values. By combining features during partitioning, it effectively addresses data bias issues and optimizes the model using ranking boosting techniques.

The CatBoost algorithm is based on BoostingTree and employs a unique approach to calculating leaf node values. It utilizes oblivious trees as the basic predictor and constructs nodes layer by layer using complete binary trees (*Chen & Han, 2021*). This approach allows for efficient storage of all leaf nodes and enables effective mining of the original data to generate new classification features. These features serve as input data for RBF neural networks, significantly improving the learning efficiency while mitigating the risk of overfitting (*Gopi, Jyothi & Narayana, 2023*). As shown in Fig. 2. The training process of the CatBoost-RBF fusion model involves feature crossover operations. In this study, GridSearch

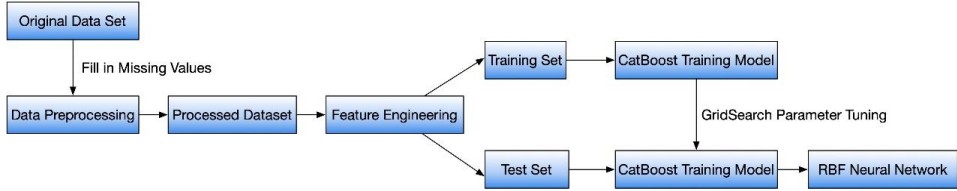

**Figure 2** **CatBoost-RBF combination process.** In this study, GridSearch is utilized to fine-tune the parameters of CatBoost, ensuring optimal performance. The leaf nodes obtained after CatBoost training are saved, and the output layer produces the probability of user repurchase behavior.

is utilized to fine-tune the parameters of CatBoost, ensuring optimal performance. The leaf nodes obtained after CatBoost training are saved, and the output layer produces the probability of user repurchase behavior.

The process of classifying and identifying abnormal online consumer behavior data involves several stages. In the first stage, feature extraction is performed on the original data. Initially, consumer features are extracted, followed by the application of principal component analysis to analyze these features. This analysis aids in reducing the initial feature set to an optimized number of features. Subsequently, a self-encoder network is utilized to compress the initial feature set, resulting in a compressed feature set. Transitioning to the second stage, the sample set that has undergone dimensionality reduction is used to construct a classification model for abnormal online consumer behavior data. The model's parameters are fine-tuned using the grid search method. The evaluation index is then employed to identify the most suitable model. Finally, the chosen model is applied to classify and identify the abnormal online consumer behavior dataset, yielding the final classification results for such data.

A self-encoder typically refers to a neural network comprising three layers. However, when performing dimensionality reduction with a self-encoder, it's crucial to ensure effective utilization of the output feature data in subsequent modeling tasks. Yet, a basic three-layer network can be sensitive to the initially set artificial values, potentially impacting the output feature data. To address this, the article employs the stacked self-encoder method, involving training the layers sequentially. The parameters learned from the self-encoding transformation between two layers are used as starting values for subsequent layers. This approach results in more stable feature data output from the network. Given the resource-intensive nature of training multi-layer neural networks and the low dimensionality of the data in this study, a stacked self-encoder structure with only two layers of encoders is adopted.

## Improved homomorphic encryption algorithm

When the traditional homomorphic encryption algorithm was proposed, there were four schemes, one of which can be expressed as follows: set two large prime numbers as p and q, let m = pq, a group of integers $Z_m$ is selected as the unencrypted space of the consumption information, the operations that can be performed on the consumption information include F = +m, −m × m. The space of encrypted consumer information is

$Z_p \times Z_q$, based on this space we can derive F is equivalent to F$'$ is equivalent. The key of the encrypted consumption information is defined as $k = (pq)$, which is expressed as a function of $Ek(a) = [a \bmod p, a \bmod p]$.

Based on the previous section's exposition on homomorphic encryption, the decryption of consumer information can be achieved through the utilization of the Chinese remainder theorem. However, the conventional homomorphic encryption algorithm, as demonstrated, fails to meet the desired standards of security concerning known consumer information. Thus, the purpose of this study is to enhance the traditional homomorphic encryption approach by employing an encryption algorithm possessing homomorphic characteristics.

When the homomorphic encryption of a consumer message is arranged in a sequential manner, it becomes arduous to withstand the impact of the encrypted consumer message. Consequently, within the scope of this investigation, we propose the following improvements to the conventional homomorphic encryption algorithm:

(1) First, set M, N (1) Firstly, set the number of large prime numbers in the network information to be secure, then let $P = M*N$;

(2) set Q represent the security parameters of the network information;

(3) the space of the end-encrypted consumer information can be expressed as $X = Y_P$. The encrypted consumption information space can be expressed as $X' = (Y_M * Y_N)^q$;

(4) Select any two prime numbers in the network information, and denote them as $D_M$ and $D_N$, and also satisfy $D_M \in Y_M, D_N \in Y_N$;

(5) Determine the encryption key of the network message, and denote the key as $K = (M, N, D_M, D_N)$;

(6) Execute the homomorphic encryption algorithm,

Set the end of the encrypted network information $M \in Z_M$, let X be randomly divided into N copies, *i.e.,* . $X_1, X_2, X_3, \ldots X_N . X_i \in Z_M, i = 1, 2, 3, \ldots, N$.

$$X = \sum_{i=1}^{N} X_i \bmod(m) \tag{2}$$

$$E_k(X) = \{[X_1 D_M \bmod M, X_1 D_N \bmod N], \ldots, [X_N D_M^N \bmod M, X_N D_N^N \bmod N]\} \tag{3}$$

where mod(m) represents the modulo operation, which divides the sum by m and takes the remainder to produce the final result.

(7) Decrypt the network information, *i.e.,*

$$D_k(X) = \begin{array}{l} [X_1 D_M D_M^{-1} \bmod M, X_1 D_N D_N^{-1} \bmod N], \\ [X_2 D_M D_M^{-2} \bmod M, X_2 D_N D_N^{-2} \bmod N], \ldots, \\ [X_N D_M D_M^{-N} \bmod M, X_N D_N D_N^{-N} \bmod N] \end{array} \tag{4}$$

where $D_M^{-N}$ and $D_N^{-N}$ are the multiplicative inverse of the corresponding powers of $D_M \bmod M$ and $D_N \bmod N$.

$$\sum_{i=1}^{N} [X_i \bmod M, X_i \bmod N] = \left[ \sum_{i=1}^{N} X_i \bmod, \sum_{i=1}^{N} X_i \bmod N \right] = [X \bmod M, X \bmod N]. \tag{5}$$

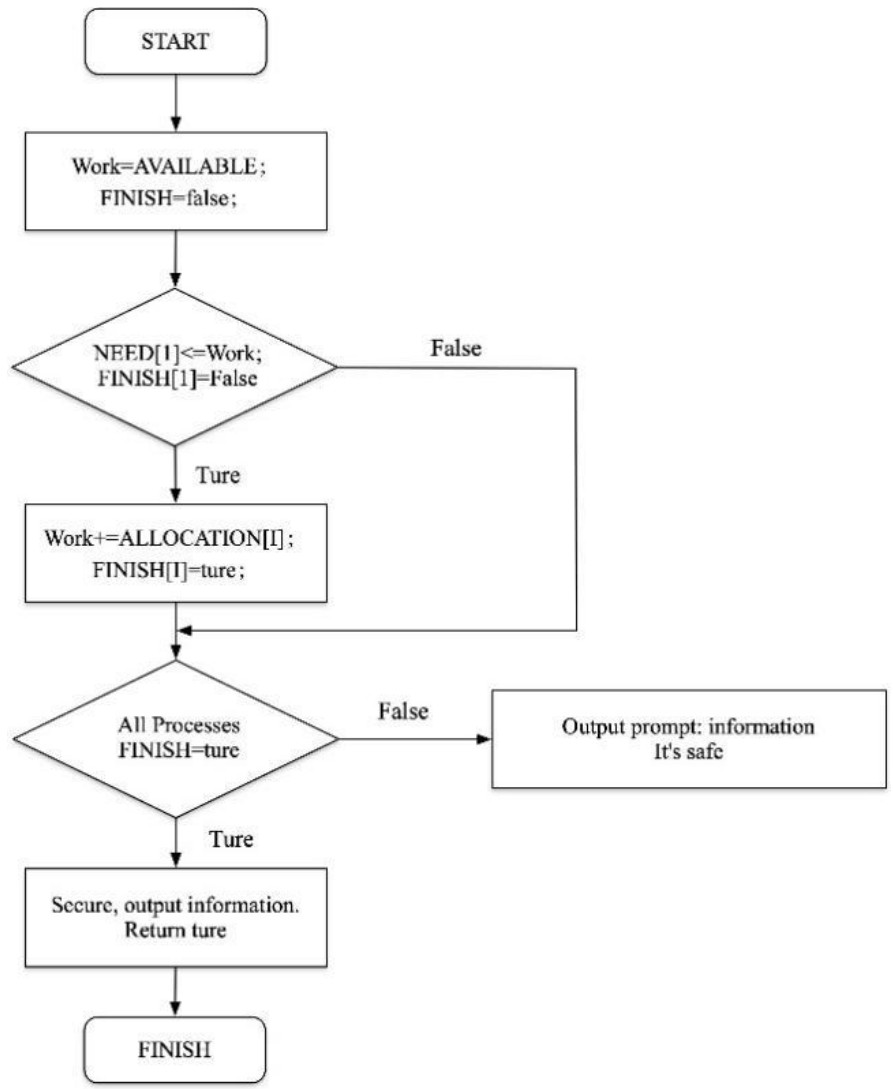

**Figure 3  Consumer data security protection process.** The process of consumer information security protection based on homomorphic encryption algorithm is shown.

Using the Chinese residual theorem, we can calculate that

$$D(X) = \left(XNN^{-1} + XMM^{-1}\right) \bmod P \tag{6}$$

where $NN^{-1} = 1 \bmod M, MM^{-1} = 1 \bmod M$.

The process of consumer information security protection based on homomorphic encryption algorithm is shown in Fig. 3.

The aforementioned procedure elucidates the manipulation of decimal values for individual data entries. Since the consumption amount is exclusively measured in yuan, angle, and cent, the number of decimal places associated with the data is restricted to a maximum of two. In practical scenarios, it is customary to establish a predetermined

number of decimal places, indicated as $n = 2$, to govern the precision of the data. Consequently, when the actual number of decimal places in a specific data entry falls below n, the remaining decimal places are supplemented with zeros. This approach is employed to facilitate accurate and efficient computations while ensuring uniformity in the representation of decimal values.

The implementation of a fixed number of decimal places, such as $n = 2$, enables standardized processing and eliminates any potential ambiguity arising from varying decimal representations. This consistency in decimal representation streamlines computational operations and enables seamless integration in various contexts. Furthermore, the inclusion of zero-fillers for decimal places below n guarantees conformity of all numerical values to a consistent format, thereby facilitating precision in subsequent calculations.

By adhering to these conventions in practical applications, the manipulation and analysis of decimal data become more streamlined and cohesive. The establishment of a defined number of decimal places, along with the implementation of zero-padding, enhances computational efficiency and minimizes the risk of errors or inconsistencies arising from inconsistent decimal precision.

# EXPERIMENTS AND ANALYSIS

## Data set

The training set in this study comprises the data from the initial nine days of an e-commerce platform, while the data from the tenth day serves as the test set for evaluation. The dataset employed in this research is sourced from the publicly available dataset provided by Alibaba Tianchi Big Data Platform, consisting of real traffic logs from Taobao's product recommendation system, dataset link is https://zenodo.org/record/8275743. After undergoing pre-processing steps such as data feature extraction and dimensionality reduction, the dataset utilized in this experiment encompasses a total of 630,008 records, with 267,652 data records representing 6,388 distinct users. Following repeated training, the model exhibits an average running time of 40 s.

## Model comparison

The selection of logistic regression (LR), LightGBM, CatBoost, and CatBoost with the radial basis function (RBF) kernel (CatBoost-RBF) for performance comparison in a machine learning context can be attributed to a strategic choice aimed at comprehensively evaluating their suitability for the task at hand. Logistic regression is included as a baseline model due to its simplicity and interpretability, allowing for a fundamental understanding of the problem. LightGBM, a gradient boosting framework, is chosen for its ability to handle large datasets and deliver high predictive accuracy through ensemble methods. CatBoost is included for its specialization in handling categorical features efficiently and its reputation for robust performance. The introduction of CatBoost-RBF, incorporating a non-linear kernel, reflects an interest in assessing whether non-linear relationships between features and the target variable significantly impact predictive performance. This selection strategy enables a holistic examination of various model characteristics, such as simplicity,

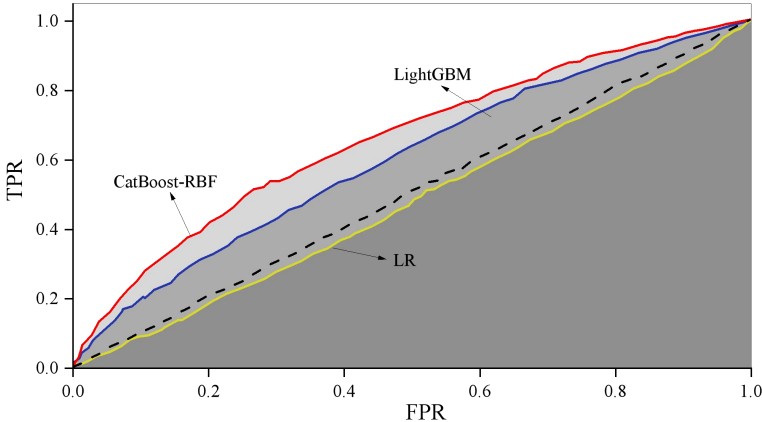

**Figure 4   ROC curve comparison.** Illustrates the comparison of training outcomes, showcasing the performance variations among different models.

accuracy, efficiency, categorical feature support, and non-linear modeling capabilities, aiding in the informed choice of the most appropriate model for the specific dataset and problem domain.

When employing logistic regression (LR) to train the original dataset, the imbalanced nature of the sample data often results in low recall scores and consequently, poor recall performance. To address this issue, data balancing is necessary, achieved through the application of appropriate weights using a weighting method. The training process is then conducted based on the adjusted results. For imbalanced datasets, the training process of the model is adjusted by assigning different weights to each class. The weights are usually inversely proportional to the frequency of the categories, that is, the minority class is weighted more and the majority class is weighted less. In this article, the weights are calculated using the inverse class frequency. Figure 4 illustrates the comparison of training outcomes, showcasing the performance variations among different models.

The combined CatBoost-RBF model demonstrates higher accuracy compared to CatBoost, LR, and LightGBM. Furthermore, the fusion model exhibits superior accuracy and precision when compared to the single RBF model. These results suggest that the CatBoost-RBF model is highly effective in predicting users' consumption behavior.

In terms of the ROC curve area AUC, the three algorithms (LR, LightGBM, and CatBoost-RBF fusion model) yield values of 0.48, 0.60, and 0.66, respectively. This indicates that the CatBoost-RBF fusion model outperforms the other two algorithms in terms of generalization performance, demonstrating better predictive value and greater generalization ability.

To validate the performance of the models, a comparison and analysis were conducted between the aforementioned models and the CatBoost-RBF classification model. Parameter tuning was carried out using a grid search method for each classification method. The results of the model and indicator measures are depicted in Fig. 5.

In Fig. 5, it is evident that except for LR, which exhibits the lowest accuracy, the other models demonstrate comparable accuracy levels. Among them, the CatBoost algorithm achieves the highest classification accuracy, indicating its superior classification performance. Regarding accuracy and recall, the random forest model displays high accuracy but low recall, suggesting its proficiency in identifying normal data but poor classification ability for abnormal data. On the other hand, the LightGBM model exhibits the highest precision rate, implying that it classifies the lowest percentage of normal data as abnormal. In contrast, the CatBoost-RBF model achieves the highest recall rate, indicating its ability to classify the lowest percentage of abnormal data as normal. In terms of F1-score, the CatBoost-RBF model exhibits the best overall performance.

Since this article primarily focuses on the classification of abnormal consumption data, with a greater emphasis on the error of classifying abnormal data as normal, the recall value assumes greater importance. Hence, the CatBoost-RBF algorithm, which performs well in both recall value and F1-score, is considered optimal. The experimental results showcase that this method attains a classification accuracy of 98.56%, precision of 97.47%, recall of 99.36%, and an F1-score of 98.41% for abnormal online shopping data. These results significantly outperform other algorithms and demonstrate superior classification recognition capabilities.

## Encryption effect

Figure 6 illustrates that the running time of the CatBoost-RBF algorithm on ciphertexts exhibits a linear increase as the data volume expands. Upon testing various datasets, the enhanced clustering algorithm requires approximately 1 s to process 2000 sets of 20-bit ciphertexts for 2D data. Furthermore, the contour coefficient achieves a level of approximately 0.4, indicating satisfactory performance. From these observations, it can be deduced that as the data volume increases, the proposed algorithm's processing time also extends. However, it is important to note that the classification effect remains consistently good and stable.

Figure 7 presents the comparison results between the privacy protection methods of the relevant schemes. In the figure, option 1 indicates the inclusion of all options, while option 2 represents their exclusion.

In the SecureML scheme, two cloud servers are required to directly reconstruct the model parameters during the training process. However, there is a potential risk of information leakage associated with this approach. On the other hand, the PPMLaas model adopts a strategy to avoid the computationally expensive bootstrap operation of homomorphic encryption. In this approach, the server checks the noise level in the ciphertext after each operation. If the noise level surpasses the threshold, the server transmits the ciphertext to the user for decryption. Subsequently, the server encrypts the modified ciphertext and transmits it back to the server.

The scheme proposed in this article ensures privacy protection for model parameters, user data, and inference results by maintaining the relevant computational data and original input data in a randomly split state during the training process. This approach restricts each cloud server involved in the computation to possess only a portion of the data. As

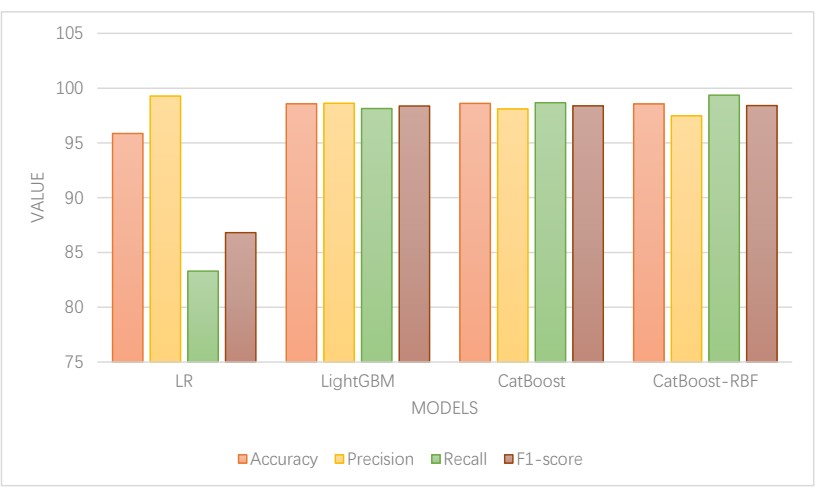

**Figure 5** **Model comparison.** Parameter tuning was carried out using a grid search method for each classification method. The results of the model and indicator measures are depicted the figure.

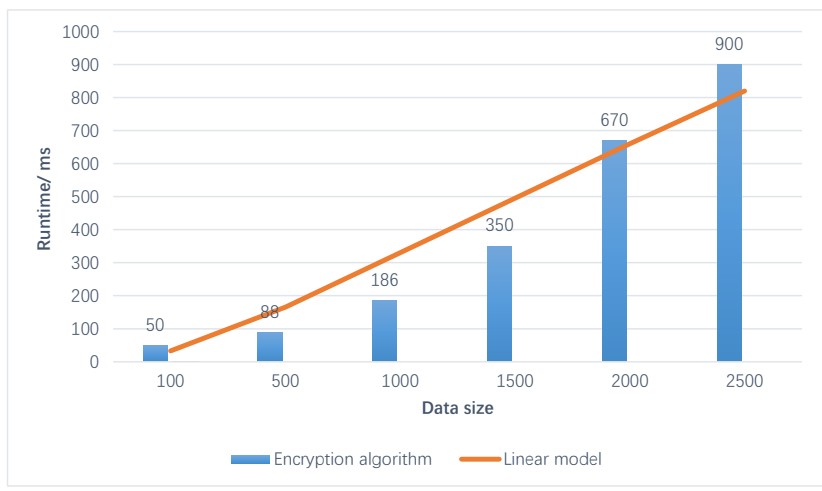

**Figure 6** **Results of run time.** From these observations, it can be deduced that as the data volume increases, the proposed algorithm's processing time also extends. However, it is important to note that the classification effect remains consistently good and stable.

a result, no single cloud server can access the complete information, thereby achieving privacy protection. In the experimental setup involving cloud servers, the strategy we adopted was to control each server to access and participate in data processing tasks. This is achieved through data sharding, which splits the synthetic dataset into smaller, more manageable partitions or shards. These partitions contain different subsets of data and form the basis for subsequent computation operations. In the experimental design, cloud servers are assigned by policy to process specific data partitions, thus ensuring that each

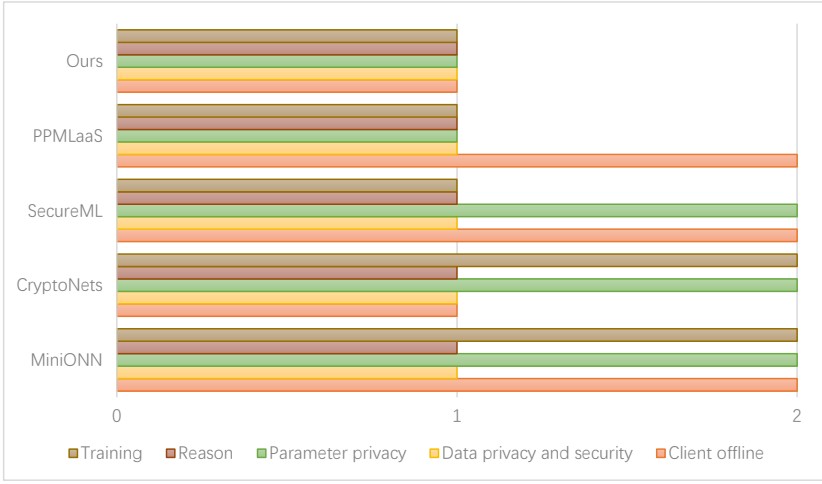

**Figure 7** **Comparison of consumer data security.** If the noise level surpasses the threshold, the server transmits the ciphertext to the user for decryption. Subsequently, the server encrypts the modified ciphertext and transmits it back to the server.

server only operates on its assigned subset of the dataset. The allocation process considers the effects of randomization, workload distribution, and server capacity.

## DISCUSSION

The analysis and research conducted on the Alibaba public dataset contribute to a profound comprehension of consumption data and abnormal behavior data across diverse business application scenarios. The formulation of online consumer behavior data anomaly detection and classification models, as presented in this article, provides significant insights for related research pursuits. The proposed model not only demonstrates advantages in consumer data classification but also effectively tackles privacy protection concerns in consumer data applications. This model facilitates the efficient and cost-effective utilization of data from various sources, thereby supporting the creation of more scientifically grounded marketing strategies built upon extensive consumer data.

In the domain of training machine learning models, a substantial challenge arises from the need for large quantities of data. However, a concerning issue has surfaced involving unethical practices such as the unauthorized collection, storage, utilization, and even trafficking of consumers' personal information and purchasing records, driven by economic motives. These actions not only infringe upon individual privacy rights but also expose consumers to potential risks of personal and financial harm. Given the paramount importance of safeguarding consumer data, the secure collection and ethical utilization of such information have become pressing priorities. Proposing privacy protection models that are specifically tailored to consumer data scenarios is imperative. Such models must ensure the preservation of data privacy, uphold security standards, and comply with regulatory mandates. At present, grappling with these challenges remains of

utmost significance in the competitive landscape of offline markets. Efforts to address these concerns are pivotal for maintaining integrity and trust within the industry.

The proposed methodology, which amalgamates consumer data and homomorphic encryption, presents a forward-looking approach to digital marketing. However, its application is not without limitations. Firstly, it introduces computational complexity due to the use of homomorphic encryption, potentially resulting in extended processing times, thereby impeding real-time or high-frequency applications. Secondly, representing CatBoost's leaf node data as input layer data can inflate data dimensions, necessitating substantial computational resources and storage capacity, which may pose challenges in high-dimensional data processing. Moreover, the security of the methodology heavily relies on key management and the robustness of the encryption algorithm, making it vulnerable to key compromises or algorithmic vulnerabilities. Additionally, employing powerful machine learning algorithms like CatBoost for hyperparameter tuning and feature extraction can be resource-intensive, potentially rendering it unsuitable for resource-constrained environments.

Nonetheless, the methodology also exhibits promising application prospects. It enables personalized marketing strategies by categorizing user online consumption behavior, enhancing user satisfaction. Furthermore, it emphasizes privacy protection through homomorphic encryption, mitigating the risks associated with handling sensitive consumption data and facilitating compliance with data privacy regulations. The secure sharing of consumer data, especially in cross-organizational collaborations or with data partners, becomes feasible through this approach, as data can be processed and analyzed without compromising the original information. Additionally, it aids in market research by providing insights into market trends and competitor behaviors, facilitating informed strategic decisions. In conclusion, while this methodology holds potential in the realm of digital marketing, its realization necessitates addressing technological and security challenges. Continuous technological advancements and improvements will likely expand its application scope. However, rigorous security measures must be upheld to safeguard the privacy of user data.

## CONCLUSION

The introduced predictive model, which amalgamates the CatBoost algorithm with the RBF neural network, undergoes extensive validation through experiments conducted on online consumer data. The GridSearch methodology is employed to determine optimal parameters for CatBoost, which are subsequently integrated with the traditional RBF neural network. The resulting model exhibits exceptional performance in forecasting user repurchasing behavior. Comparative analysis of various models verifies that the CatBoost-RBF amalgamation attains the highest classification accuracy, underscoring its effectiveness in predicting consumer purchasing behavior. By introducing this integrated approach, the study effectively addresses the pivotal challenge of developing precise predictive models in the realm of consumer data analysis. A solitary cloud server proves insufficient for acquiring comprehensive data, thus preserving privacy concerning model parameters, user data, and

inference results. The obtained results validate the proposed methodology, highlighting its potential to provide valuable insights for marketers, optimizing their strategies, and deepening their understanding of consumer behavior. Nonetheless, further exploration is imperative to investigate additional avenues for refining and expanding the proposed model. This endeavor will contribute to enhancing its applicability and broadening its scope within the field of consumer data analytics.

## ACKNOWLEDGEMENTS

We would like to thank the anonymous reviewers whose comments and suggestions helped improve this manuscript.

### Funding
The authors received no funding for this work.

### Competing Interests
The authors declare there are no competing interests.

### Author Contributions
- Jun Cui conceived and designed the experiments, performed the experiments, performed the computation work, prepared figures and/or tables, authored or reviewed drafts of the article, and approved the final draft.
- Hao Jiang conceived and designed the experiments, analyzed the data, prepared figures and/or tables, and approved the final draft.
- Zhendan Xu performed the experiments, performed the computation work, authored or reviewed drafts of the article, and approved the final draft.

### Data Availability
The data is available at Zenodo: None. (2023). consumer behavior data [Data set]. Zenodo. https://doi.org/10.5281/zenodo.8275767.

The code is available in the Supplementary File and at Zenodo: None. (2023). code for Digital marketing program design based on abnormal consumer behavior data classification and improved homomorphic encryption algorithm. Zenodo. https://doi.org/10.5281/zenodo.8275743.

### Supplemental Information
Supplemental information for this article can be found online at http://dx.doi.org/10.7717/peerj-cs.1690#supplemental-information.

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
