# Peer review of "Digital marketing program design based on abnormal consumer behavior data classification and improved homomorphic encryption algorithm"

_PeerJ Computer Science, doi:10.7717/peerj-cs.1690_

## Round 0.1 · original submission · Minor Revisions

Dear author
Your paper has been reviewed by the experts in their field and you will see that they have some concerns about the current quality of the manuscript. They have also suggestions for improvement. I do agree with their opinions.

Moreover please improve the abstract and justify the novelty of the work

Reviewer 1 ·

Basic reporting

The paper provides references and sufficient domain background/context. But there are still some places to improve. This paper aims to explore the design of a digital marketing scheme that relies on consumer data and homomorphic encryption. The approach involves using GridSearch to align and store the leaf nodes obtained after training the CatBoost model. These leaf node data are then used as input for the Radial Basis Function (RBF) layer, where the mapping of leaf nodes to the hidden layer space occurs. This process results in the classification of user online consumption data in the output layer. Moreover, a refinement is introduced to the traditional homomorphic encryption algorithm to enhance privacy preservation during consumption data processing. This enhancement expands the scope of homomorphic encryption to include rational numbers. The Chinese remainder theorem is integrated to decrypt consumption information. Empirical results demonstrate the exceptional generalization performance of the fusion model, showcasing an AUC (Area Under the Curve) value of 0.66, a classification accuracy of 98.56% for online consumption data, and an F1-Score of 98.41. Should the authors explain the reason for using these four models (LR, LightGBM, CatBoost, CatBoost-RBF)? The figure 1 and 2 are a bit blurry; the figures quality should be improved. Conclusions should be rewritten based on the results and discussion.

Experimental design

In addition to these four methods, whether the author has tried other methods, and if the effect of other methods is not good, it should also be shown as a comparison.

Validity of the findings

The limitations and application prospects of the method should be given.

Reviewer 2 ·

Basic reporting

This paper aims to explore the design of a digital marketing scheme based on consumer data and homomorphic encryption, which includes using GridSearch to align and store the leaf nodes obtained after training the CatBoost model. The final experimental results show that the fusion model has excellent generalization performance, but the author still needs to correct the following deficiencies to improve the paper.
1. Keywords that summarize the article are missing in the abstract;
2. The last paragraph of the introduction should be followed by columns with contributions;
3. Formula (2) to formula (6) in this paper lacks corresponding explanations;
4. Decimal consistency simplifies the calculation, and the author needs to explain the following process by which it enables seamless integration of context;
5. How to solve the problem of sample data imbalance in Section 4.2? Please give a detailed description;

Experimental design

6. If the parameters of each classification method are tuned in the experiment, does it mean that the parameters are the same?
7. In the experiment, the approach is how to limit each cloud server involved in the computation to owning only a portion of the data?

Validity of the findings

8. The conclusion seems to be similar to the abstract, lacking a summary of the article's content and the prospect of the future.

Additional comments

NA

---

## Round 0.2 · accepted · Accept

Dear authors

Thanks for your resubmission, based on the input from the experts, I'm pleased to inform you that your manuscript has been accepted for publication.
Thanks for your fine contribution

Reviewer 1 ·

Basic reporting

No changes required

Experimental design

authors address all the points

Validity of the findings

authors address all the points

Reviewer 2 ·

Basic reporting

All the concerns have been addressed. The paper can be accepted in its current state.

Experimental design

N/A

Validity of the findings

N/A